# Context-Aware Chatbot Extension Leveraging HTML Data and Retrieval-Augmented Generation (RAG)

**Yang Ouyang**
Department of Computer Science and Technology
Tsinghua University
ouyy22@mails.tsinghua.edu.cn

**Tong Yu**       **Wenchu Wang**
Department of AI Music and Music Information Technology
Central Conservatory of Music
tongyu@mail.ccom.edu.cn
wenchu@mail.ccom.edu.cn

## 1   Introduction

In the digital age, the need for efficient information retrieval has become increasingly crucial. With the vast amount of information available on the web, users often find themselves in situations where they are seeking quick answers or additional information related to the current page they are viewing. However, the process of switching between multiple pages or looking up external resources can be disruptive and time - consuming, which significantly affects the user experience.Traditional chatbots, while being useful to some extent, often lack the ability to adapt to the context. Specifically, they struggle to offer responses directly based on a user's past and present interactions with a website. This shortcoming has led to a demand for more intelligent information retrieval systems.

To address these issues, the objective of this project is to develop a browser - based chatbot extension. This chatbot will be designed to retrieve and synthesize information from two main sources: previously parsed HTML data and traditional Retrieval - Augmented Generation (RAG) sources. By leveraging both of these sources, the chatbot aims to provide accurate, context - sensitive responses. Moreover, it will be powered by a fine - tuned LLaMA 3 model [1]. This model, which is capable of handling HTML context, is expected to enhance the user experience by providing contextual insights and enabling more efficient information retrieval. In essence, this project bridges the gap between the user's need for quick and relevant information and the limitations of existing chatbot technologies by integrating stored HTML context and external data sources, thus offering users streamlined access to relevant information through a browser extension. This development is not only important for improving the user experience in day - to - day web browsing but also has the potential to impact the field of information retrieval and natural language processing, setting new standards for context - aware chatbot systems.

## 2   Definition

When constructing browser-based chatbot, the similarity calculation formula of query and HTML data is of great significance. When the query content is entered by the user and converted to vector $\vec{q}$, its similarity to the pre-parsed HTML data vector $\vec{H}$ is calculated. Provide accurate basic information for subsequent processing, and then improve the accuracy of chatbot response.

$$sim(\vec{q}, \vec{H}) = \frac{\vec{q} \cdot \vec{H}}{|\vec{q}||\vec{H}|}$$

38th Conference on Neural Information Processing Systems (NeurIPS 2024).

Take the LLaMA 3 model and fine-tune it. When combining information obtained from HTML data and traditional RAG sources, the model makes predictions about this information, resulting in a probability distribution $p = (p_1, p_2, \cdots, p_N)$ while there is a true label distribution $y = (y_1, y_2, \cdots, y_N)$. The loss function quantifies the difference between the prediction and the real result, and provides the basis for the optimization algorithm to adjust the model parameters. The model is constantly optimized, and the prediction is constantly close to the real results, thus ensuring that the chatbot's response to user queries is constantly improved in accuracy and rationality, and ensuring the overall response quality.

$$L = -\sum_{i=1}^{N} y_i \log(p_i)$$

## 3   Related Work

In the digital era, the proliferation of web information has posed numerous challenges to traditional information retrieval. As the amount of data on the web has grown exponentially, users often encounter difficulties in obtaining the information they need efficiently. For instance, seeking additional information related to the current page often requires cumbersome and time - consuming multi - page - switching, degrading user experience. Traditional chatbots, while somewhat useful, lack the ability to adapt to user - interaction contexts and struggle to generate responses relevant to users' past and present website interactions [2]. Retrieval - Augmented Generation (RAG) [5] is an innovative NLP [3] approach that combines retrieval and generation to answer complex questions. Recently, Lee et al. [4] proposed SPARC, which uses contextualized representations to encode term interactions and capture word meanings. Combining these with sparse features improves question - answering performance.

## 4   Proposed Method: Dual-Source Retrieval-Augmented Generation (RAG) Chatbot

1. **Preprocessing and Indexing**
   - **Database**: Build and index a database from previously parsed and stored web content, allowing quick retrieval of relevant historical information and supporting context continuity across user sessions.
   - **Web Data**: Crawl and index additional unstructured data (e.g., reviews, discussions) for broader context, using embeddings to enable semantic search.
2. **Query Analysis and Retrieval Decision**
   - **Intent Classification**: Identify whether the query is specific (favors database), general (requires both database and additional web data).
   - **Source Selection**: If the query is specific, prioritize retrieval from the historical web content database; for general queries, retrieve from both sources.
3. **Retrieval and Consolidation** Retrieve relevant data from both sources, if applicable, and structure it into a prompt for the generative model.
4. **Response Generation** Feed the structured prompt (query + retrieved data) into a fine - tuned LLaMA 3 model to generate a coherent response.
5. **Feedback Loop** Collect user feedback to refine query classification, retrieval thresholds, and model fine - tuning for improved accuracy and relevance.

## 5   Expected Outcomes

- An intelligent chatbot extension that can pull directly from parsed HTML data, providing users with responses rooted in both current and historical webpage context.
- Enhanced accuracy and contextual relevance in responses due to integration with stored HTML and traditional RAG sources.
- A scalable solution that supports ongoing user interactions across various web pages and provides accurate, up-to-date information from both the local webpage and external sources.

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
