# OpenReview forum: "Context-Aware Chatbot Extension Leveraging HTML  Data and Retrieval-Augmented Generation (RAG)"
_tsinghua.edu.cn/THU/2024/Fall/AML — THU 2024 Fall AML Submission_

### Official Review · ~Yinuo_Li1 · 2024-11-06
**Good idea but not enough related works reviewed**

**Rating:** 8
**Confidence:** 4

**Review:**

This proposal has a very clear definition about their problems and goals, and has a solid statement about the significance of achieving their goals. The proposal also has a detailed approach method and their expected outcomes which showed their careful thinking on this problem.

However, the related works reviewed is very limitied and too general, lack of detailed review on different sub-topics. For example a lot of AI search applications came out recently which may have somehow solved a part of the challenges they addressed in the proposal which said LLM can hardly handle the massive information online.

---

### Official Review · ~Iat_Long_Iong1 · 2024-11-07
**Great idea despite improvement required**

**Rating:** 8
**Confidence:** 5

**Review:**

This research proposal outlines an innovative approach to developing a context-aware chatbot extension. It aims to enhance the accuracy and relevance of chatbot responses, particularly in the context of web browsing, by integrating HTML data with Retrieval-Augmented Generation (RAG) techniques.

The proposed chatbot extension faces the challenge of processing lengthy HTML contexts, potentially exceeding 500,000 characters. A general HTML simplification scheme is crucial for its success. Additionally, comparing its efficiency with existing web-based systems like WebGPT and WebGLM would strengthen the research.


Minor Typo:
- Hyphens are used without spaces, e.g., “day - to - day” -> “day-to-day”.

---

### Official Review · ~Lei_Wu17 · 2024-11-09
**Evaluation of "Context-Aware Chatbot Extension Leveraging HTML Data and Retrieval-Augmented Generation (RAG)"**

**Rating:** 9
**Confidence:** 4

**Review:**

# Pros:
* Innovative Use of HTML Parsing with RAG: Introducing HTML parsing alongside traditional RAG sources is an original approach that could set a new standard for chatbot interactions.
* Clear Structure and Methodology: The paper’s structured approach aids comprehension, with defined steps from query analysis to response generation.
* Potential for Broad Application: This work has potential applications in various domains where contextual relevance is crucial, offering scalability across different web environments.
# Cons:
* Limited Implementation Details: While the methodology is outlined, there is a lack of specific implementation insights, such as the HTML parsing and fine-tuning challenges with LLaMA 3.

---

### Official Review · ~Sui_Yuanpei1 · 2024-11-10
**Good concept, but some technical complexities may impact implementation and scalability**

**Rating:** 8
**Confidence:** 4

**Review:**

This proposal is centered on the development of a context-sensitive chatbot capable of retrieving and synthesizing information from two sources: stored HTML data and RAG sources. The project addresses the limitations of traditional chatbots in contextual awareness and web-based information retrieval, which is both timely and relevant given the increasing complexity of web interactions.

Pros:
1.The dual-source RAG model combined with fine-tuning a powerful LLaMA 3 model is a strong choice for generating accurate, contextually relevant responses.
2.The chatbot prioritizes user experience by reducing the need for users to switch between multiple pages, which can improve interaction efficiency and satisfaction.
3.By indexing parsed HTML data and unstructured external data, this system can potentially handle a variety of web contexts and adapt to evolving user needs.
4.Implementing a feedback loop shows a commitment to continuous improvement, allowing the chatbot to refine its accuracy and relevance over time.

Cons:
1.The proposed integration of HTML parsing, RAG, and LLaMA 3 fine-tuning may require significant computational resources and technical expertise, which could complicate implementation.
2.Given the use of stored web data, there may be privacy considerations, especially if user interactions and browsing history are stored.
3.The accuracy of the chatbot's responses is heavily dependent on the fine-tuning quality of the LLaMA 3 model, which may require substantial trial and error.
4.While the system may work well within controlled environments, it is uncertain how effectively it will perform across a broad range of websites and complex data structures.

---

### Official Review · ~Zihan_Yan2 · 2024-11-10
**A good and clear proposal**

**Rating:** 9
**Confidence:** 3

**Review:**

This proposal aims to develop a browser-based chatbot extension that leverages HTML data and Retrieval-Augmented Generation (RAG) technology to provide users with accurate, context-sensitive responses. The proposal clearly explains the background and significance of the problem and holds extensive application potential.

---

### Official Review · ~Zou_Dongchen1 · 2024-11-10
**Review of this proposal**

**Rating:** 8
**Confidence:** 4

**Review:**

In this proposal, the authors propose to build a browser - based chatbot extension that quickly retrieves and extracts information such as relevant HTML based on the user's query and returns the results to the user. This solves a real-world pain point problem: users always have to switch between different web pages to query available information, but existing large models have not yet addressed this difficulty.
I find this proposal to be practical and actionable. The authors have clearly stated the expected method to conduct this project. However, recently ChatGPT has gone live with a search function, which means that it searches the web in real time based on user input and generates answer messages accordingly. I’m not quite sure what are the advantages of the author’s proposed method when compared to the existing solutions on the market.

---

### Official Review · ~Liutao7 · 2024-11-10
**The proposal has clear practical value and is also a hot research direction on the Internet.**

**Rating:** 8
**Confidence:** 4

**Review:**

The proposal presents an extension of a context-aware chatbot that combines HTML data with RAG technology, which is innovative and practical. However, there are some technical details that need refinement, and it is necessary to further clarify the similarities and differences with existing solutions on the market.

Advantages:

Completeness: The proposal outlines the project's objectives, background, methods, and technical details, including steps such as preprocessing, indexing, query analysis, retrieval decision-making, response generation, and feedback loops, which is relatively comprehensive.
Creativity: The project combines HTML data with RAG technology, providing a clear solution for the extension of context-aware chatbots, with practical application value.
Workload: The project involves multiple technical fields, requiring fine-tuning of models and annotation of data, indicating a substantial workload.

Areas for improvement:

Clarify the evaluation system: Define the methods for evaluating the accuracy and context relevance of the chatbot, demonstrating certain improvements over traditional retrieval methods.
Clarify scalability: The size and quality of the dataset have a significant impact on the final results. If applied to complex real-world internet environments, further adjustments and optimizations may be needed.

---

### Official Review · ~Junjie_Chen1 · 2024-11-11
**Good and Clear Proposal**

**Rating:** 8
**Confidence:** 4

**Review:**

The proposal presents a clear structure and addresses an important problem of improving chatbot systems by incorporating context-awareness through dual-source Retrieval-Augmented Generation (RAG). The methodology is well-thought-out, leveraging both pre-parsed HTML data and external RAG sources to enhance response relevance and accuracy. The use of a fine-tuned LLaMA 3 model and the feedback loop further showcases the feasibility of the approach and its potential for scalability.

However, the baselines for comparison are not well-detailed, which could limit the assessment of the proposed system's relative performance.

---

### Official Review · ~Zihan_Lv1 · 2024-11-11
**Clear methodology and expected results**

**Rating:** 8
**Confidence:** 3

**Review:**

The project is innovative in terms of context-awareness and combining RAG technology and the overall technical architecture is clear. High practical value in improving users' web browsing experience could be seen. For further application, large-scale scenarios, especially under complex web structure and high concurrent requests, which is still to be verified.

---

### Official Review · ~Tianhai_Liang1 · 2024-11-11
**Good Proposal**

**Rating:** 8
**Confidence:** 4

**Review:**

This proposal aims to develop a browser-based context-aware chatbot extension leveraging HTML data and Retrieval-Augmented Generation (RAG) with a fine-tuned LLaMA 3 model. It combines stored webpage content with external sources to generate accurate, context-sensitive responses, enhancing user information retrieval and experience.

The system offers real-time, context-aware responses by leveraging current HTML data, reducing page-switching for users. Integrating both HTML and RAG sources enhances response accuracy. It's scalable and adaptive, using a fine-tuned LLaMA 3 model for coherent, relevant answers.

The chatbot's accuracy depends on the quality of stored HTML data. Fine-tuning LLaMA 3 is resource-intensive, potentially limiting small teams. Accessing user webpage data raises privacy concerns.

---

### Official Review · ~Gangxin_Xu1 · 2024-11-12
**Review of "Context-Aware Chatbot Extension Leveraging HTML Data and Retrieval-Augmented Generation (RAG)"**

**Rating:** 9
**Confidence:** 4

**Review:**

This proposal presents a browser-based chatbot extension powered by a fine-tuned LLaMA 3 model, which aims to enhance information retrieval through context awareness. By utilizing both parsed HTML data and Retrieval-Augmented Generation (RAG) techniques, the chatbot seeks to provide relevant, context-sensitive responses based on a user’s interactions with a webpage. The project intends to streamline web browsing by reducing the need for users to navigate away from pages to seek related information, thus improving the user experience and advancing context-aware chatbot technologies.

Strengths:
Innovative Use of HTML Data with RAG: Combining HTML context with RAG is a promising approach to improving chatbot contextual awareness. This structure enables more specific and relevant responses tailored to the user’s browsing environment.
Clear User-Focused Objective: The focus on minimizing user disruption by enabling in-page responses directly related to the user’s current context aligns well with improving web navigation efficiency.
Potential for Broader Impact: The project could set new standards in chatbot capabilities within information retrieval and context awareness, with applications that extend beyond casual browsing to include customer support and educational contexts.